# Mechanical Properties, Wettability and Thermal Degradation of HDPE/Birch Fiber Composite

**DOI:** 10.3390/polym13091459

**Published:** 2021-04-30

**Authors:** Agbelenko Koffi, Fayçal Mijiyawa, Demagna Koffi, Fouad Erchiqui, Lotfi Toubal

**Affiliations:** 1Centre de Recherche sur les Matériaux Lignocellulosiques, Université du Québec à Trois-Rivières, Boul. des Forges, C.P. 500, Trois-Rivières, QC G9A 5H7, Canada; agbelenko.koffi@uqtr.ca (A.K.); faycal.mijiyawa@uqtr.ca (F.M.); demagna.koffi@uqtr.ca (D.K.); 2Laboratory of Biomaterials, University of Quebec at Abitibi-Témiscamingue, 445, Boul. de l’Université, Rouyn-Noranda, QC J9X 5E4, Canada; fouad.erchiqui@uqat.ca

**Keywords:** compounding, composite, birch fiber, mechanical properties, thermogravimetric analysis, dynamic mechanical analysis

## Abstract

Wood–plastic composites have emerged and represent an alternative to conventional composites reinforced with synthetic carbon fiber or glass fiber–polymer. A wide variety of wood fibers are used in WPCs including birch fiber. Birch is a common hardwood tree that grows in cool areas such as the province of Quebec, Canada. The effect of the filler proportion on the mechanical properties, wettability, and thermal degradation of high-density polyethylene/birch fiber composite was studied. High-density polyethylene, birch fiber and maleic anhydride polyethylene as coupling agent were mixed and pressed to obtain test specimens. Tensile and flexural tests, scanning electron microscopy, dynamic mechanical analysis, differential scanning calorimetry, thermogravimetry analysis and surface energy measurement were carried out. The tensile elastic modulus increased by 210% as the fiber content reached 50% by weight while the flexural modulus increased by 236%. The water droplet contact angle always exceeded 90°, meaning that the material remained hydrophobic. The thermal decomposition mass loss increased proportional with the percentage of fiber, which degraded at a lower temperature than the HDPE did. Both the storage modulus and the loss modulus increased with the proportion of fiber. Based on differential scanning calorimetry, neither the fiber proportion nor the coupling agent proportion affected the material melting temperature.

## 1. Introduction

To meet the growing demand for durable construction materials of natural origin, a new class of product known as wood plastic composites (WPCs) has emerged. As production techniques and chemical modification methods improve, WPCs are becoming more and more commonplace. Compared to conventional composites reinforced with synthetic carbon fiber or glass fiber–polymer, WPCs represent a lower density, lower cost, and recyclable alternative [1,2,3,4]. According to a report by Grand View Research Inc., the global demand for wood-fiber-reinforced plastic composites is projected to increase by 144% from $4.46 billion in 2016 to $10.89 billion by 2024 [5]. However, while the use of some natural fiber composite materials expands in industries such as construction, sports equipment and automotive parts, their development remains limited due to the lack of knowledge about their behavior and sensitivity to environmental factors, especially in the case of plant-based short fibers [3,6,7,8,9,10]. 

In many applications, metal gears have been replaced by plastic gears because, and despite their many benefits, the intensive use of plastics raises sustainability issues because of the pollution that is generated. Developers of new materials can no longer ignore the impact that these will have on the environment. Reinforcement with short natural fibers is proposed as a way of enhancing the mechanical properties of low-cost plastics and limiting their environmental impact. Wood increases the strength and stiffness of thermoplastics at minimal cost and has been the preferred filler in the plastics industry. The mechanical properties of wood fiber composites appear not to be affected significantly by wood type [11], although it has been reported that the type of lignocellulosic fiber and the lignin, cellulose and hemicellulose content have strong influences on the mechanical properties of the material [12]. Due to its ease of handling and widespread availability, wood flour is the most used lignocellulosic filler. Fiber morphology, including the length and length/diameter ratio, affects the mechanical properties of WPCs [13,14,15,16]. Thermoplastics used with lignocellulosic materials must melt or soften at or below the thermal degradation point of the lignocellulosic component, normally 200–220 °C [17]. Polypropylene, polystyrene, polyvinylchloride, low-density polyethylene and high-density polyethylene all meet this requirement. However, satisfactory dispersion of wood filler in a thermoplastic matrix has long been a challenge due to chemical incompatibility. Wood fiber is hydrophilic whereas plastic is hydrophobic. In order for the two phases to share stress loads, the interface between them must be extensive and durable. To increase the affinity between the two components, the surface properties of at least one of them must be modified. The compatibility of wood and plastic can be improved by introducing a compatibilizer into the system. By virtue of its bifunctionality, the compatibilizer is assumed to increase adhesion between wood and plastic. Maleated polyethylene (MAPE) increases the tensile and flexural strengths of the resulting composite material [18,19,20]. In composites made with polyethylene, it is the most widely used compatibilizer [21,22]. A relatively small proportion (2–3% by weight) is sufficient to optimize interfacial stress transfer [21,23,24,25,26]. 

A wide variety of wood fibers are used in WPCs. Two popular choices are aspen and birch. Although both have been studied in depth, they have not been compared in terms of WPC mechanical and thermal properties. The tensile and impact properties of linear low-density polyethylene (LLDPE) reinforced with up to 30% aspen fiber coupled with a silane [27] and up to 40% coupled with a titanate [28] have been studied. Birch-fiber-reinforced polyethylene composite has been studied in terms of creeping [29], tensile properties [13,30], compatibilized interfacial adhesion and effects of recycling on mechanical properties [31,32,33]. Birch is a common hardwood tree that grow in cool areas with abundant precipitation, such as the province of Quebec, Canada [29,34].

In addition to material mechanical properties, wettability and thermal degradation as a function of wood fiber content must be studied. Although wood fiber is hydrophilic and hence easily wetted with water, its affinity for water can change once it is incorporated into a hydrophobic thermoplastic such as polyethylene. However, the main challenge with wood fiber is its degradation during composite manufacturing at temperatures in the 200–230 °C range [3]. The thermal degradation of aspen fiber has been studied, but without considering the degradation of the composite [17]. Degradation is associated primarily with the breakdown of components such as cellulose, hemicellulose and lignin to low molecular weight substances. 

Polyethylene (PE) is one of the most widely used thermoplastics in the world. Its toughness, near-zero moisture absorption, excellent chemical inertness, low coefficient of friction, high dielectric constant, very low electrical conductivity and ease of processing make it suitable for many material applications, such as pipes and tubing, sheets, containers and electrical insulation of wiring and cables [35,36]. The mechanical and physical properties of PE depend significantly on variables such as the type and degree of branching, crystallinity and average molecular weight. Composites made with PE as the matrix are used widely in applications requiring physical and mechanical properties that are weaker in the pure polymer. PE is sometimes mixed with another polymer to improve the mechanical properties of the composite. With the addition of 5 wt% PTFE in HDPE, tensile strength, Young’s modulus and impact strength of in situ nanofibrillar composites were about 6.0, 6.0 and 3.1 times higher than that of neat HDPE, respectively [36]. PE composites are used in packaging, electrical components, thermal energy storage, automotive parts, biomedical and aerospace applications [3,35,36,37].

In this work, we studied the tensile and flexural properties, wettability and thermal degradation of a composite material developed using birch fiber with MAPE and high-density polyethylene (HDPE).

## 2. Experimental

### 2.1. Preparation of the Composite Material

#### 2.1.1. Materials

High-density polyethylene with a melt flow index of 20.0 g/10 min and density of 0.962 g/cm³ was provided by NOVA Chemicals Inc. Yellow birch wood pulp fiber was prepared at the Institute of innovations in eco-materials, eco-products and energy-efficient biomass (I2E3) of the University of Quebec in Trois-Rivières (UQTR) according to the thermomechanical extraction method. The thermomechanical pulping process (TMP) was used to make the fibers. This process uses wood chips subjected to a temperature above 100 °C in steam in order to soften the fibers. Pressurized defibration ensues in a refiner fitted with two rotating discs rotating at high speed. It is the effect of successive cycles of compression and decompression that produces the dough at a yield rate of approximately 90%. Wood fiber was dried at 80 °C in an air-circulating oven for 24 h and then ground to 20–60 mesh size before use. The fiber aspect ratio (mean length divided by average diameter) classes were obtained by mechanical refining and screening and characterized using an OpTest fiber quality analyzer (Table 1).

Maleated polyethylene provided by the Eastman Chemical Company contained 1.5% maleic anhydride and had a molecular weight of 15,000 (MAPE G2010).

#### 2.1.2. Compounding

The materials were prepared by blending the components in a Thermotron mixer (C.W. Brabender, model T-303, Figure 1). High density polyethylene (HDPE) and Maleated polyethylene (MAPE) at a mass ratio of 20:3 were melted on rollers at 170 °C. Wood fiber and the remaining HDPE were then blended in for 7 min at 60 rpm. The blend was peeled off the roller and re-blended five times for 3 min each to obtain a uniform composite sheet, which was removed from the roller and cut into strips with a knife to fit into the samples mold. Blends without coupling agent were also prepared. The fiber content ranged from 0 to 50% by weight at increments of 10%.

#### 2.1.3. Compression Molding

Strips of composite material were molded into dog-bone shaped specimens (ISO 527-2:2012 Type 1A) for tensile strength testing (Figure 1) [38] and into rectangular shape for the three-point flexural test ASTM D790-10 [39]. A DAKE press was used (Figure 1). The metal plates were heated on both sides at 170 (±2) °C. The pressure was maintained at 5 MPa for 10 min (although fiber degradation begins around 250 °C, applying 5 MPa of pressure for a long time causes fiber degradation). The mold was then cooled to below 60 °C by circulating cold water in the plates. The dimensions of the specimens were 10 mm wide and 4 mm thick for the tensile test and 12.2 mm wide and 3.1 mm thick for the flexural text. The production process is shown schematically in Figure 1. All specimens were polished, and then kept overnight in the testing room. The width and thickness were measured with a micrometer.

### 2.2. Mechanical Characterization

Tensile and flexural tests were conducted on an Instron device (Model LM-U150) at room temperature. For the tensile test, the Instron was equipped with a 150 kN load cell, calibrated in a range of 0 to 50 KN. A 25 mm extensometer connected to the data acquisition system was fixed to the gauge length section of the specimen to record strain. The testing speed was 2 mm/min. The tensile properties, including strength at maximum load, modulus of elasticity and strain at break, were determined according to ISO 527-1:20 [40]. Six replicates were tested for each composite.

Because of sample flexibility, a 10 kN load cell was used for the flexural test in order to increase accuracy. The support span was 55 mm, which produced a span-to-depth ratio of 16 ± 1. The crosshead speed *V* was 1.5 mm/min, which was calculated as follows:(1)$$V=\frac{ZL^{2}}{6d}\;$$
where *Z* (mm/mm/min) is the strain on the outer fiber, *L* (mm) is the support span and *d* (mm) is the sample depth. In ASTM method D790, *Z* = 0.01 mm/mm/min for procedure A [39].

### 2.3. Surface Energy Measurement

The WPC samples were conditioned at 65% RH and 20 °C. The surface energy was calculated by measuring the contact angles of droplets of water and bromonaphthalene with the solid surface using a camera connected to a desk-top computer. The FTA32 video software processes the outline of the droplet and then determines the contact angle using interpolation methods. The surface energy is then estimated using the Zisman and Owen-Wendt methods [41] with the Owens-Wendt Equation (2):(2)$$(1+\cos\theta )\;\gamma_{L}=2\left[{\left(\gamma_{L}^{d}\gamma_{S}^{d}\right)}^{1/2}+{\left(\gamma_{L}^{p}\gamma_{S}^{p}\right)}^{1/2}\right]$$
where θ is the contact angle of the sample with the test liquid, $\gamma_{L}^{d}$,$\;\gamma_{L}^{p}$ and $\gamma_{L}$ are respectively the dispersive, polar and total surface energies of the test liquid and $\gamma_{S}^{d}$, $\gamma_{S}^{p}$ are, respectively, the dispersive and polar components of the surface energy of the tested solid material. The total surface energies of the liquids used are 72.8 $\mathrm{mJ}/m^{2}$ for water and 44.4 $\;\mathrm{mJ}/m^{2}$ for bromonaphthalene.

### 2.4. Thermogravimetric Analysis (TGA)

The thermal decomposition of the composite materials was studied using a thermogravimetric analyzer (TGA 8000, PerkinElmer, Waltham, MA, USA) as per ASTM D 2584. Samples (virgin HDPE, birch fiber, 10, 30, 50% wood fiber by weight composite) weighing 5–10 mg were scanned over a temperature range of 50–575 °C at a rate of 5 °C/min under nitrogen. To avoid the influence of the pressure, the composites were taken from the sheets obtained after mixing. The differential weight loss was recorded as a function of temperature and the corresponding initial and final decomposition temperatures were recorded.

### 2.5. Scanning Electron Microscopy

A scanning electron microscope (JSM 5500, JEOL, Boston, MA, USA) was used at an acceleration voltage of 12 kV to check sample surfaces for the presence of fiber after compounding and molding. These observations were carried out in the I2E3 forestry materials lab at UQTR.

### 2.6. Dynamic Thermomechanical Analysis

A Q series dynamic mechanical analysis (DMA) (TA Instruments, New Castle, DE, USA) provided by Waters LLC, USA was used with specimens 30 mm long × 10 mm^2^ rectangular cross-section mounted in a dual cantilever clamp. The temperature range was 20–150 °C and the stress load was applied at frequencies of 1–20 Hz. The composite viscoelastic performance was evaluated in terms of storage modulus, loss modulus and tan δ. At least two measurements were performed on material of each composition.

### 2.7. Differential Scanning Calorimetry (DSC)

A Q series DSC (TA Instruments) was provided by Waters LLC, USA. About 10 mg of sample sealed in an aluminum pan were heated from 20 °C to 200 °C at a rate of 10 °C/min. The sample was cooled to 20 °C then heated again to 200 °C. The procedure was carried out under nitrogen. Melting temperature of HDPE/birch fiber composites, based on differential scanning calorimetry was measured [42].

## 3. Results and Discussion

### 3.1. Mechanical Characterization

#### 3.1.1. Tensile Test

Figure 2 shows the tensile stress–strain curves of HDPE composites containing 10–50% wood fiber by weight. Strain at failure decreased while maximal stress and material stiffness increased with fiber content. Stiffness means that the molecules are less mobile and therefore decreases the tolerated strain (Figure 2).

Tensile strength, elastic modulus and strain at break values of HDPE/birch fiber composite made with 3% MAPE are reported in Table 2. The strength and the modulus increased linearly with the fiber weight fraction, with strong correlations in both cases. This tensile modulus behavior has been reported previously for HDPE reinforced with hardwood [43], whereas tensile strength was found to behave otherwise [44], peaking at 25% (vol.) 40-mesh hardwood fiber and decreasing as the proportion of 20-mesh fiber increased. However, this was in the absence of a coupling agent. In our case, the tensile modulus increased by 212% as birch fiber content increased from 0 to 50% by weight, while tensile strength increased by 93% (Table 2). The significant increases in the tensile strength and modulus are due not only to the fiber content, but also to better adhesion between fiber and matrix [18,19,20,21,22] because of the coupling agent.

The elastic modulus of our compression-molded HDPE composite containing 40% birch fiber was comparable to that of similar material (including the fiber length to width ratio of 20) made previously by injection molding [30]. For other HDPE/hardwood composite bodies produced by compression molding, the elastic modulus was the same as in the present experiment even as fiber content varied [44].

#### 3.1.2. Bending Test

The effect of birch fiber content on the flexural stress–strain curves of the composite material is shown in Figure 3. The maximal stress endured and the stiffness both increased with fiber content.

Mechanical failure occurred in materials containing 30% or more birch fiber. Up to this concentration, deformation reached the maximum possible under the test conditions (a support span of 55 mm). Beyond 20% fiber, the material broke while the strain at break decreased. Increasing fiber content reduces the mobility of molecules in the polymer matrix and, therefore, decreases strain. The same behavior was observed in the tensile test, presumably for the same reason.

Flexural strength, flexural modulus and strain at break values of HDPE/birch fiber composite made with 3% MAPE are reported in Table 2. Like the tensile properties, the flexural strength and flexural modulus increased linearly with the fiber weight fraction. The correlations were strong (based on R^2^) in both cases. Similar behavior has been reported for HDPE reinforced with eucalyptus fiber and bamboo fiber and 5% coupling agent [19]. In the present study, the flexural modulus increased by 234% as the birch fiber content increased from 0 to 50% (Table 2) and ended up higher than that of HDPE [45]. This pattern of behavior was observed previously in wood-reinforced composites of HDPE [19] and of low-density polyethylene [46].

While birch fiber increased the tensile strength by 180% (Table 2), the flexural strength of increased by nearly three-fold. Other researchers studying other types of wood fiber fillers have obtained similar results [16,47,48].

### 3.2. Wettability and Surface Energy

The contact angles measured for the test liquids on the surface of the composite materials containing different amounts of birch fiber are shown in Table 3. These results show that the two calculation methods show the same pattern for the effect of fiber content on the total surface energy and give similar total energy values. Since birch fiber is a more polar substance than HDPE, it might be expected that composite material containing more fiber would be more hydrophilic and therefore have a higher surface energy. However, fiber increased the polar component of the surface energy by very little. The total surface energy of the material was due primarily to the dispersive component of HDPE [49]. This suggests that even on the surface, the fibers remain coated by the matrix, which the SEM images appear to confirm (Figure 4). The visible difference between the surfaces of the matrix (Figure 4a) and of the composite containing 50% fiber (Figure 4b) is slight.

### 3.3. Thermogravimetric Analysis

Figure 5 illustrates the thermal stability of virgin HDPE and birch fiber composites between 50 °C and 600 °C.

The loss of HDPE mass occurred in a single-stage degradation process that occurred over the temperature range of 400–500 °C, giving a single peak in the first-derivative curve with a DTG_max_ of 475 °C (Table 4). Birch fiber by itself decomposed at temperatures between 220 °C (T_d_) and about 400 °C, with a shoulder near 270 °C and the weight loss rate peaking at about 360 °C. Plant polymers of lower molecular weight, such as hemicellulose, degrade in the 225–325 °C range [15,18], whereas cellulose degrades in the 300–400 °C range [15]. Lignin decomposes in a wide temperature range (200–600 °C) [50]. In contrast, the thermal degradation of HDPE/wood composites occurred in a two-step process (Table 4), which is confirmed by the presence of two peaks on the first-derivative plots, with maxima at nearly the same temperature for the three wood fiber contents tested. Initial weight loss thus may have corresponded to charring of the hemicellulose, cellulose and lignin constituents of the wood fiber, followed by breakdown of the HDPE in the second step. The thermal stability of the composites is thus intermediate between those of birch fiber and of HDPE. Adding birch fiber thus decreases the thermal stability of HDPE, since its thermal stability is lower overall [51].

Although temperature does not determine the degradation rate of the fiber, the thermal stability of the composite material tends to decrease as the birch fiber content increases. In other words, the thermal stability of the material tends to be that of the wood fiber filler. This behavior was observed elsewhere [52,53].

### 3.4. Dynamic Mechanical Analysis

DMA results showed the viscoelastic properties of the resin and the composites. As shown in Figure 6 and Figure 7, the storage modulus decreases as a function of temperature and regardless of the fiber content, at both 1 Hz and 20 Hz frequencies of stress loading. The material thus loses its strength as it is heated, which is attributable to the easing of movement of polymer chain segments as the surrounding thermal energy increases. This property of wood fiber plastic composites has been noted previously by several authors [54,55]. At both frequencies, the storage modulus increases significantly as a function of fiber content, thus confirming the reinforcing role of the fiber. The 61% storage modulus increase from 2300 MPa (virgin HDPE) to 5900 MPa (HDPE50B) at 30 °C, 1 Hz on addition of fibers is due to the increase in stiffness of composites compared to the resin. Finally, although the storage modulus appears to have increased slightly when the frequency increased from 1 Hz to 20 Hz, this effect was not significant.

Figure 8 and Figure 9 show that the loss modulus E” increases, reaches a maximum and then decreases as a function of temperature, regardless of the fiber content and at 1 Hz and 20 Hz. This behavior was observed previously in HDPE reinforced with wood fiber [56].

At both frequencies, the loss modulus increased significantly as a function of fiber content. The role of the fiber as a reinforcer is again apparent. Finally, regardless of birch fiber content, the loss modulus was slightly lower at 20 Hz than at 1 Hz.

Figure 10 and Figure 11 show that, regardless of fiber content, the damping ratio tan delta increased with temperature, again at both 1 Hz and 20 Hz. This behavior has been noted previously in HDPE reinforced with wood fiber [57]. At both frequencies, the damping ratio decreased with increasing wood fiber content, due mainly to the decreased mobility of HDPE molecules. Finally, the damping ratio decreased when the stress application frequency increased.

### 3.5. Differential Scanning Calorimetry

Figure 12 shows that neither birch fiber nor the coupling agent and coupling agent concentration affect the melting temperature of the composite material. All melting temperatures (HDPE and composites) are between 137 °C and 145 °C. HDPE’s glass transition Tg is around −120 °C; and with experimental conditions (heating from 20 °C to 200 °C), the influence of fibers and fiber content in composites at Tg cannot be studied. Crystallization of the composites which could affect the mechanical properties of the resulting composites was not investigated in this study but in another study parallel to this one.

## 4. Conclusions

The production method used in this study yielded HDPE/birch fiber composite material having tensile and flexural properties with excellent reproducibility and a low standard deviation. The proportion of filler affected measurably the mechanical properties, wettability and thermal degradation of the material. Thermal properties were assessed by DSC and TGA. Results show that regardless of fiber content, the damping ratio tan delta increased with temperature, again at both 1 Hz and 20 Hz. At both frequencies, the damping ratio decreased with increasing wood fiber content, due mainly to the decreased mobility of HDPE molecules. Finally, the damping ratio decreased when the stress application frequency increased. DSC shows that neither birch fiber nor the coupling agent affects the melting temperature of the composite material. Conclusions drawn from this work are as follows:The elastic modulus, the storage modulus, the loss modulus and the ultimate strength increase as a more or less linear function of the fiber content.The elastic modulus can be increased by over 200%, the flexural strength by 180%.The tensile strength and the flexural modulus are indifferent to the fiber content.Birch fiber has little impact on the material surface energy; the matrix remains the principal determinant of the surface properties.Birch fiber degrades at a lower temperature than does HDPE, giving rise to two peaks on the composite material thermal breakdown curves. The mass loss at a given temperature increases with the initial fiber content.Neither the birch fiber content nor the coupling agent content affects the melting temperature of the material.

## Figures and Tables

**Figure 1 polymers-13-01459-f001:**
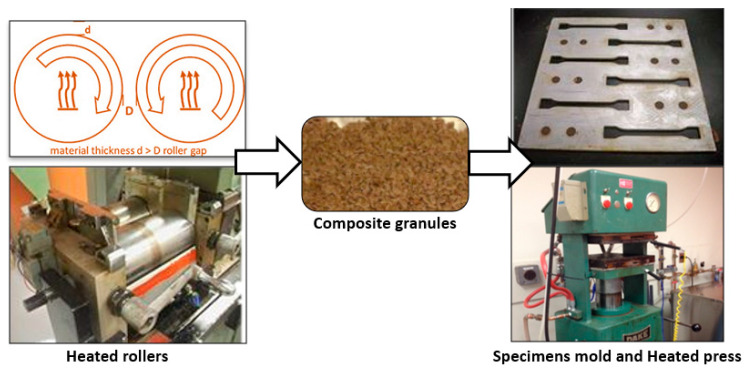
Summary of production process for composites and samples.

**Figure 2 polymers-13-01459-f002:**
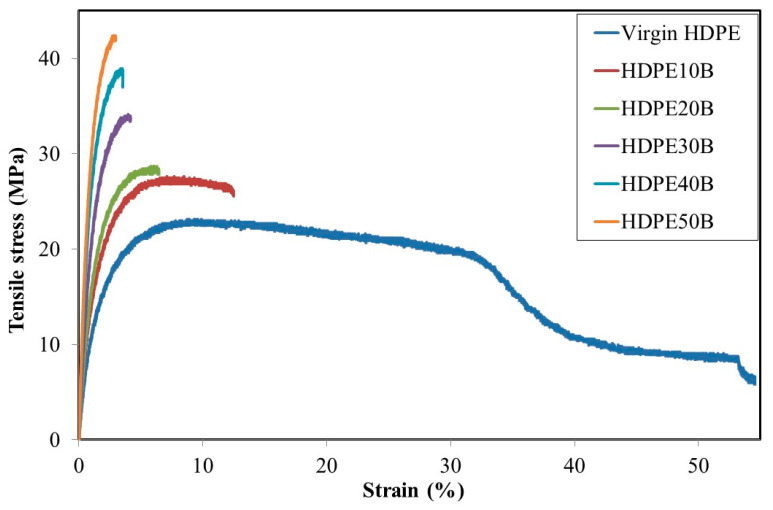
Tensile test: stress–strain behavior of HDPE/birch fiber composites.

**Figure 3 polymers-13-01459-f003:**
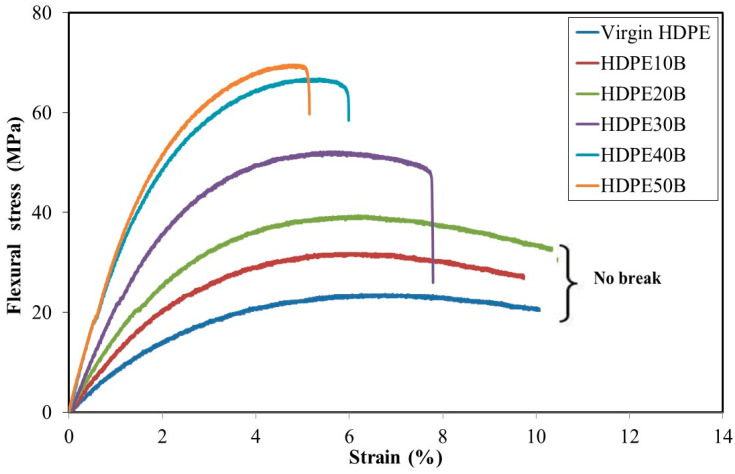
Flexural test: stress–strain behavior of HDPE/birch wood composites.

**Figure 4 polymers-13-01459-f004:**
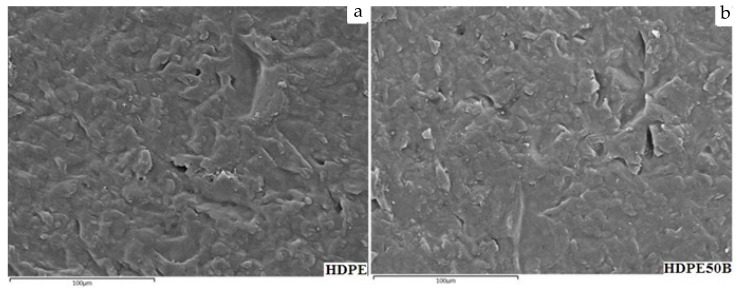
SEM of (**a**) virgin HDPE and (**b**) HDPE containing 50% birch fiber.

**Figure 5 polymers-13-01459-f005:**
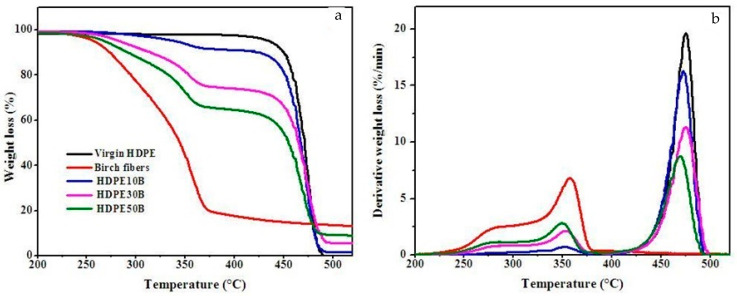
(**a**) Thermogravimetric curves and (**b**) first-derivative analysis of the thermal breakdown of virgin HDPE, birch fiber and HDPE/fiber composites.

**Figure 6 polymers-13-01459-f006:**
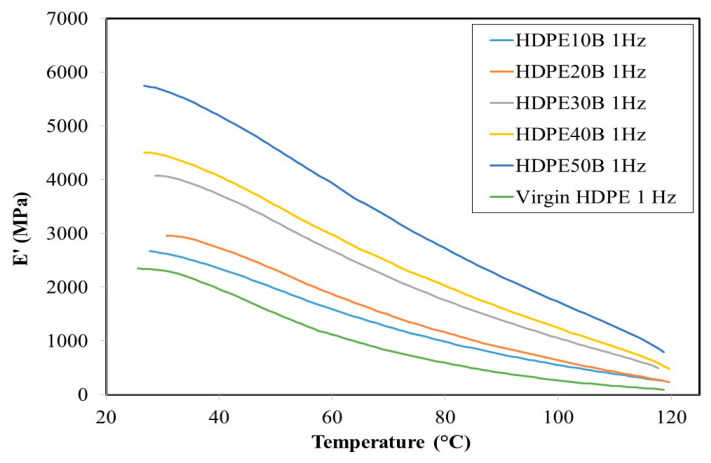
Storage modulus of HDPE/birch fiber composites at 1 Hz.

**Figure 7 polymers-13-01459-f007:**
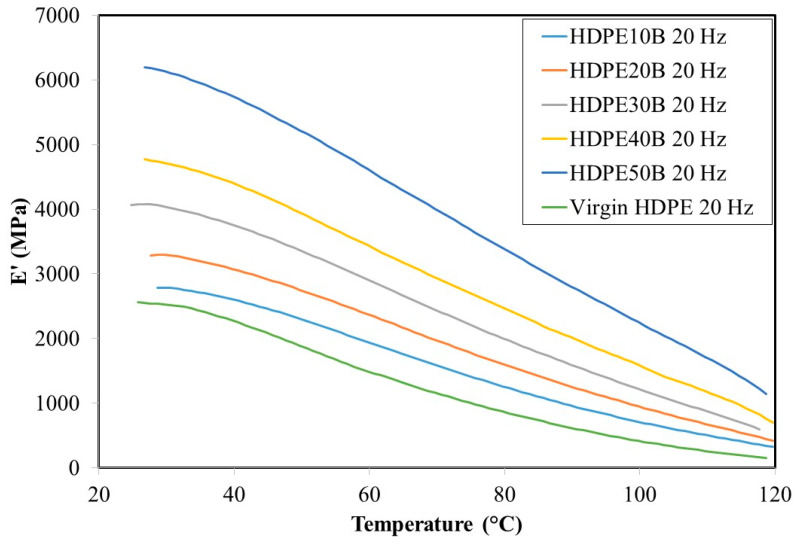
Storage modulus of HDPE/birch fiber composites at 20 Hz.

**Figure 8 polymers-13-01459-f008:**
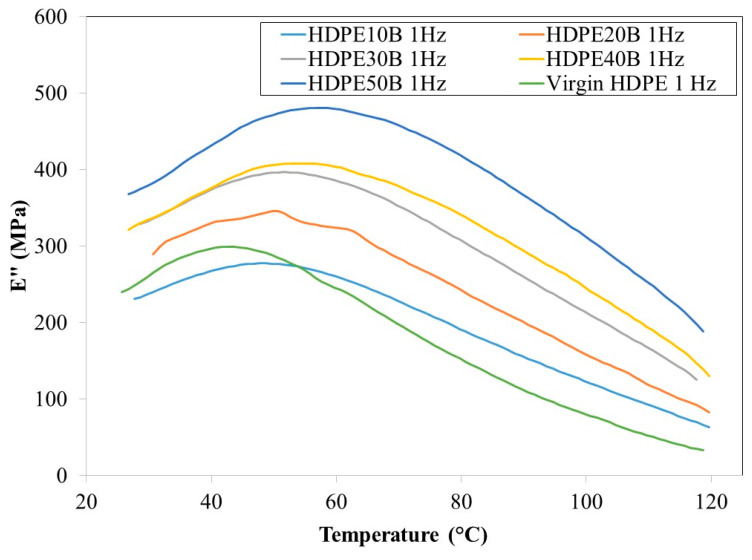
Loss modulus of HDPE/birch fiber composites at 1 Hz.

**Figure 9 polymers-13-01459-f009:**
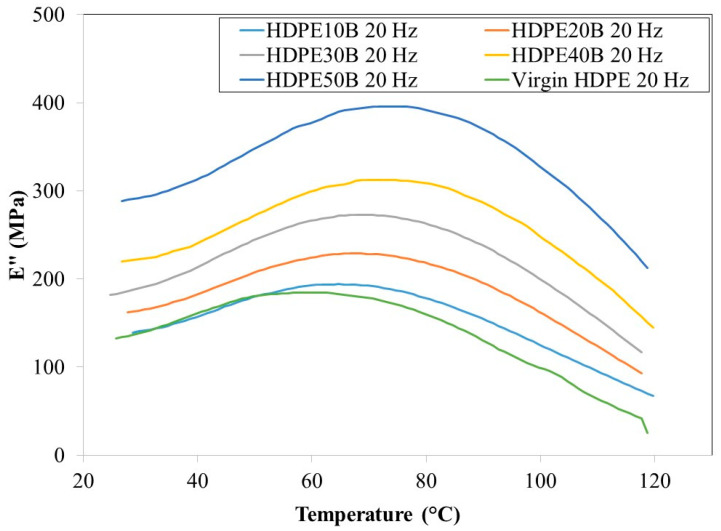
Loss modulus of HDPE/birch fiber composites at 20 Hz.

**Figure 10 polymers-13-01459-f010:**
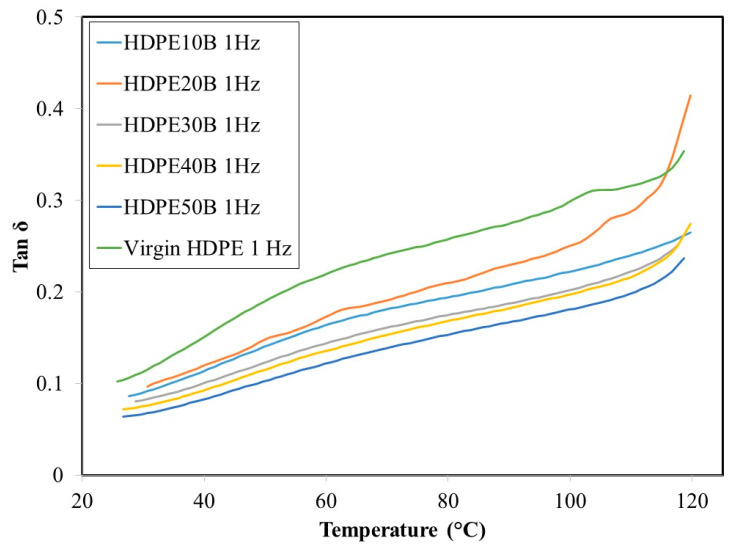
Temperature dependency of the damping ratio tan ό of HDPE/birch fiber composites at a stress frequency of 1 Hz.

**Figure 11 polymers-13-01459-f011:**
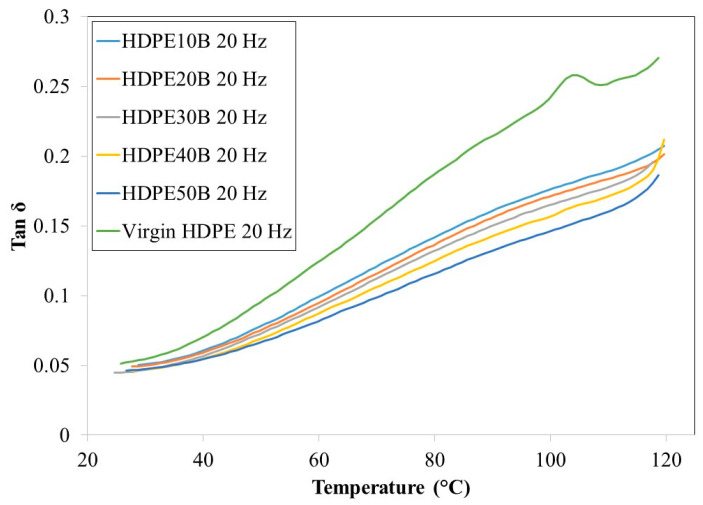
Temperature dependency of the damping ratio tan ό of HDPE/birch fiber composites at a stress frequency of 20 Hz.

**Figure 12 polymers-13-01459-f012:**
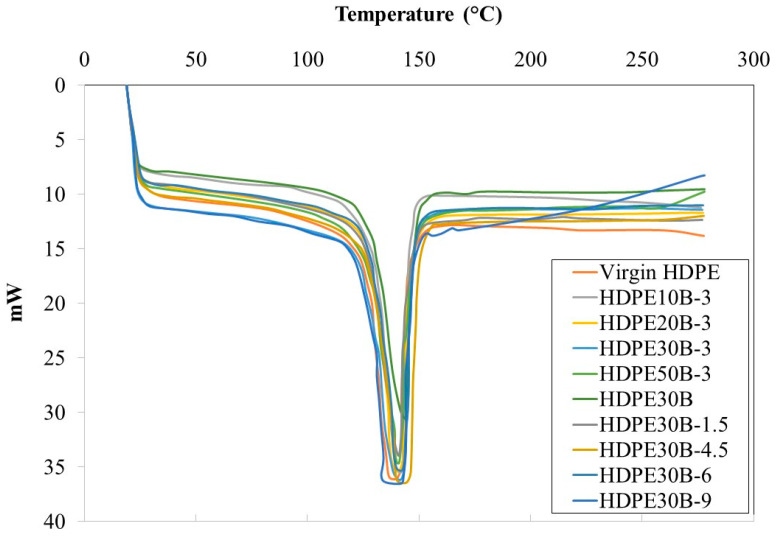
Melting temperature of HDPE/birch fiber composites, based on differential scanning calorimetry; the number preceding ‘B’ indicates the fiber concentration, the number following indicates the coupling agent concentration.

**Table 1 polymers-13-01459-t001:** Birch fiber characterization properties.

Dimension	Measured
Mean length: L (mm)	0.49
Mean width: D (µm)	24.7
Aspect ratio: L/D	19.79
Fiber count	5000

**Table 2 polymers-13-01459-t002:** Mechanical properties of HDPE/birch fiber composite material.

Nomenclature	Fiber Content (%)	Tensile Properties			Flexural Properties		
		Modulus (GPa)	Strength (MPa)	Strain at Break (%)	Modulus (GPa)	Strength (MPa)	Strain at Break (%)
Virgin HDPE	0	1.34 (±0.06)	22 (±0.85)	No break	1.04 (±0.16)	23.90 (±0.93)	No break
HDPE10B	10	1.71 (±0.05)	27.07 (±0.52)	10.76 (±1.88)	1.39 (±0.09)	33.19 (±2.93)	No break
HDPE20B	20	1.83 (±0.44)	28.53 (±0.48)	6.60 (±0.24)	1.99 (±0.19)	42.40 (±2.15)	No break
HDPE30B	30	2.7 (±0.10)	34.98 (±1.70)	3.85 (±0.57)	2.38 (±0.26)	50.49 (±1.12)	8.23 (±0.53)
HDPE40B	40	3.45 (±0.33)	38.05 (±2.83)	3.57 (±0.037)	3.22 (±0.32)	63.83 (±2.01)	5.44 (±0.54)
HDPE50B	50	4.19 (±0.40)	42.65 (±3.2)	2.76 (±0.46)	3.47 (±0.41)	66.70 (±3.58)	4.95 (±0.23)

Mean (±standard deviation).

**Table 3 polymers-13-01459-t003:** Contact angle and surface energy of HDPE/birch composites.

Nomenclature	Fiber Content (%)	Contact Angle (°)		Surface Energy (mJ m^−2^)			
		Water	Bromo	Owens-Wendt			Zisman
				Disp.	Polar	Total	Total
Virgin HDPE	0	110.70 (±2.05)	41 (±0.00)	33.90	0.00	33.90	38.11
HDPE10B	10	100.00 (±2.24)	27.50 (±1.12)	39.52	0.01	39.53	42.37
HDPE20B	20	106.60 (±1.96)	31.80 (±2.28)	37.83	0.00	37.83	40.65
HDPE30B	30	101.00 (±1.41)	33.00 (±1.87)	37.53	0.01	37.54	39.95
HDPE40B	40	84.8 (±1.92)	33.50 (±1.80)	37.33	2.45	39.78	38.05
HDPE50B	50	92.2 (±1.24)	38.30 (±1.89)	35.36	1.03	36.39	36.97

In parentheses: standard deviation; bromo: bromonaphthalene.

**Table 4 polymers-13-01459-t004:** Thermal decomposition kinetics of HDPE, birch fiber and composites.

Nomenclature	Material	First Peak (°C)	Second Peak (°C)	Fiber R_max_ (%/s)	HDPE R_max_ (%/s)
Virgin HDPE	HDPE	None	475	-	0.33
B	Birch fiber	360	None	0.14	-
HDPE10B	HDPE + 10 *	350	470	0.03	0.29
HDPE30B	HDPE + 30 *	353	476	0.04	0.19
HDPE50B	HDPE + 50 *	360	475	0.05	0.15

R_max_ = peak rate of mass loss; * = % birch fiber.

## Data Availability

The data that support the findings of this study are available from the corresponding author, [Lotfi Toubal], upon reasonable request.

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
