# Peer review of "Mechanical Properties, Wettability and Thermal Degradation of HDPE/Birch Fiber Composite"

_polymers, 2021, doi:10.3390/polym13091459_

Round 1

Reviewer 1 Report

  1. Abbreviations should not be used in an abstract. Authors have used n abbreviation MAPE without defining it. Eve the abbreviations of the common characterization techniques are not supposed to be used in an abstract.
  2. “while tensile strength increased by 93% (Error! Reference source not found.)”…There are many such instances in the manuscript. Please check this and rectify.
  3. The authors have conducted a very systematic research in terms of experiments and presented the results very systematically. However, they failed to give any sound scientific reasoning behind these good results with evidence. So, I strongly suggest that the authors should improve the discussion part in the manuscript.
  4. As they have mentioned, that dispersion of the fibers within the polymer matrix is a challenge and is one of the most important factors affecting the overall properties of the composite, they should provide some proof of good dispersion of the fiber within the polymer matrix. Moreover, they have mentioned at numerous instances the reason as good adhesion and dispersion without any evidence.
  5. I guess, they should provide the evidence in terms of SEM images of the cross-sectional fractured composites to make their argument strong.
  6. The other concern that I have is, how did the authors come up with a value of 3% of MAPE agent as the best one. Did they optimize this? If yes, then the results have to be mentioned.
  7. The authors have mentioned that the mechanical properties improved until 50% of wood fiber concentration. So, can that be extrapolated that if the fiber concentration increased further, the mechanical properties will increase. Some proof should be given about this.

Author Response

To: Mr. Stevan Stojanović

Assistant Editor

Polymers

Manuscript ID: polymers-1177802

Title: Mechanical properties, wettability and thermal degradation of HDPE/birch fiber composite

By: Agbelenko Koffi et al.

Dear Editor,

I would like to thank reviewers for their time and interest into this study. The submitted article (polymers-1177802) had been revised and enhanced with the guidelines provided by the reviewers. Thanks again for your comments that have surely improve the quality of our paper.

In the present document, one will find the corrections proposed by the reviewers.

Each comment is numbered, and some are split into sub-comments for simplicity reasons.

Strikethrough text is the text that is removed, and underlined text is the text that is added.

Reviewer 2 Report

In this work, the mechanical properties, wettability and thermal degradation of HDPE/birch fiber composite was studied. The author found that the addition of birch fiber could be improved the tensile elastic modulus, flexural modulus, storage modulus and loss modulus. Meanwhile, the composites still exhibited hydrophobic based on the water contact angle result. Indeed, the topic is within the scope of Polymers, and the results are partially supported by the experimental data. Many points needed to be solved before considering it to be accepted for publication. The detail comments are listed as follows:

-Whole Manuscript. The current version included several comments in the right side. It seems not like a final version for the submission. Please double check it.  

-Introduction section. The following papers about high-performance HDPE-based composites should be included to show your grasp of the achievements in the field. Composites Science and Technology, 2021, 207, 108715; Composites Science and Technology, 2019, 175, 100-110. In addition, the novelty of this work is not clear. Please describe it more clearly in the revised manuscript.

-Experimental section. The resolution of Figure 1 is very low. Please replace a high-resolution figure in the revision.

-Results and Discussion section. What are the mechanical properties of the PP/functionalized cellulose composites? I think that the mechanical properties of the composites is important for the use of them in practical. 

-Mechanical characterizaiton section. Though the addition of birch fiber could be improved the tensile strength and flexural strength of HDPE, the elongation at break was significantly decreased. How about the impact strength of the prepared composites?

-Page 7. Several references were missing.

-Wettability and surface energy. There were no obvious differences on the SEM images between HDPE and HDPE/50% birch fiber. It was not very convincing. Please double-check it again.

-Thermogravimetric analysis. Based on the TGA results, no positive effect of birch fiber on the HDPE composites could be found. Therefore, the addition of birch fiber only improved the tensile and flexural strength. No other benefit could be obtained. The author should be explained it.

-Whole Manuscript. Some language and grammar errors need to be corrected before publication.

Author Response

(The authors gave the same response as above.)

Round 2

Reviewer 1 Report

Thanks for answering all the queries.

Reviewer 2 Report

Accept